# Composition, Organoleptic Characteristics, Fatty Acid Profile and Oxidative Status of Cow’s Milk and White Cheese after Dietary Partial Replacement of Soybean Meal with Flaxseed and Lupin

**DOI:** 10.3390/ani13071159

**Published:** 2023-03-25

**Authors:** Stella Dokou, Antonios Athanasoulas, Stylianos Vasilopoulos, Zoitsa Basdagianni, Eleni Dovolou, Ioannis Nanas, Katerina Grigoriadou, Georgios S. Amiridis, Ilias Giannenas

**Affiliations:** 1Laboratory of Nutrition, School of Veterinary Medicine, Faculty of Health Sciences, Aristotle University of Thessaloniki, 54124 Thessaloniki, Greece; 2Hellenic Dairies SA, 5th km Trikalon-Pylis, 42100 Trikala, Greece; 3Laboratory of Animal Husbandry, Department of Animal Production, School of Agriculture, Aristotle University of Thessaloniki, 54124 Thessaloniki, Greece; 4Laboratory of Reproduction, Department of Animal Science, University of Thessaly, 41434 Larissa, Greece; 5Department of Obstetrics and Reproduction, Veterinary Faculty, University of Thessaly, 43100 Karditsa, Greece; 6ELVIZ Hellenic Feedstuff Industry S.A., 59300 Platy, Greece; 7Institute of Plant Breeding and Genetic Resources, Hellenic Agricultural Organization—DEMETER, 57001 Thermi, Greece

**Keywords:** dairy cows, soybean meal, flaxseed, lupins, antioxidant status, milk fatty acids, white cheese, organoleptic characteristics

## Abstract

**Simple Summary:**

In most Mediterranean countries, soymeal is an imported and economically unstable feedstuff that unpredictably affects the profitability of dairy farms. Here we examined whether the partial substitution of soymeal with a mixture of locally produced flaxseed and lupins could affect milk yield and cheese production in dairy cows. Sixty Holstein cows were fed either a standard food, or a meal in which 50% of soymeal was replaced by a flaxseed-lupin (FL) mixture. Milk yield and composition, as well as cheese characteristics of five replicates, were examined. The feed modification did not affect the general health and milk yield of the animals. Milk yield and composition, as well as cheese characteristics, were examined. The feed modification did not affect the general health and milk yield of the animals. Milk from the FL group had decreased levels of saturated fatty acids, while cheese made from milk from this group had increased polyunsaturated fatty acids when compared to controls. These results imply that replacement of soymeal with flaxseed and lupin is profitable as it confers special characteristics to milk and cheese that are of benefit to human health.

**Abstract:**

The effect of partial substitution of soybean meal by equal quantities of flaxseed and lupins in diets of Holstein dairy cows and heifers was investigated. A total of 6 animals (30 multiparous and 30 primiparous) were allocated into two equal groups in a randomised block design and fed control (group CO) or modified (group FL) TMR diets from three weeks prior to calving until day 40 postpartum. The TMR of group CO contained corn, barley, soybean meal, rapeseed cake, corn silage, and Lucerne hay, whereas in group FL equal quantities of whole flaxseed and lupins were used to replace 50% of the soybean meal in the TMR. All animals were fed twice daily with a daily allowance of 24 kg dry matter intake per animal. Milking was carried out three times daily and milk yield was recorded during every milking. Milk samples were analysed for chemical composition and SCC content. White cheeses were manufactured from bulk milk of each group at industrial level. Bulk milk and white cheese were analysed for chemical composition and fatty acid profile; cheese was also assessed for its organoleptic properties. Results indicate that milk yield did not differ among groups. Lipid oxidation values were similar among the groups, for both milk and cheese. However, FL inclusion resulted in lower (*p* < 0.05) protein carbonyls and higher (*p* < 0.05) phenolic compounds in both milk and cheese samples. Milk from the FL group had decreased palmitic (*p* < 0.05) and myristic (*p* < 0.05) and increased oleic (*p* < 0.05) and linolenic acid (*p* < 0.05) when compared to group CO. White cheese from group FL showed a decrease in saturated fatty acids (SFA) (*p* < 0.05), an increase in monounsaturated fatty acids (MUFA) (*p* < 0.05), and a higher increase in polyunsaturated fatty acids (PUFA) (*p* < 0.05) when compared with that of group CO. The white cheese of cows fed diets with flaxseed and lupins showed compositional and organoleptic properties quite similar to control group cheese; aroma, texture, and color were acceptable and desirable in both cheeses. However, increased levels of n-3 polyunsaturated fatty acids were found in the cheese of FL fed animals. The substitution of soybean meal by flaxseed and lupins in diets of Holstein cows warrants further investigation, especially towards the production of cheese that meet the consumers’ demand for novel and healthier dairy products.

## 1. Introduction

The nutritional value of milk and dairy products is crucial for human health throughout different stages of their life [1]. Despite their important role in human nutrition, nutritional guidelines recommend reducing the consumption of saturated fatty acids (SFA) that originate from milk and dairy products [2]. The feasibility to modify cow’s milk fatty acid (FA) profile through dietary modifications has been widely explored in the past; diets enriched with beneficial fatty acids were proven to favorably alter milk FAs, leading to higher levels of polysaturated fatty acids (PUFA) and monosaturated fatty acids (MUFA) [3,4,5]. 

Several studies have explored the effects of flaxseed incorporation in dairy cows’ diets. In Italy, it was found that feeding Holstein cows with flaxseed during the summer period increases both milk fat content and yield together with the protein and total casein yield [5]. Those results suggest that flaxseed supplementation during the summer months improves the composition and nutritional properties of the milk. In addition, feeding flaxseed to dairy cows improves the FA profile and the CLA content of milk [3]. Thus, this nutritional strategy can contribute to the improvement of the health properties of milk by enhancing CLA content, suggesting that its consumption benefits human health [3]. Similarly, milk yield, fat, and lactose, as well as feed efficiency, are improved in cows fed with 3% flaxseed oil; this is consistent with the hypothesis that milk and milk components would respond to increased energy intake [6]. White lupin is an annual legume with a rich protein content and high levels of unsaturated fatty acids, when compared to other proteinaceous feedstuffs that can be incorporated in dairy cows’ diets [7].

Besides milk, white cheese, produced from Holstein cow milk, is a popular dairy product with a meaningful nutritional role that can serve as an excellent source of high value proteins, whey, and minerals [8]. At the same time, dairy products, including white cheese, have received negative criticism, primarily due to their FAs profile, identified to be rich in SFA [9]. Cheese composition is primarily affected by milk composition, which is used as a raw material for its production. Several studies have explored the alteration of milk, and thus cheese, FAs profiles by introducing feeds rich in n-3 and PUFA in cows’ rations [3,10]. Flaxseed is an oilseed rich in n-3 fatty acids linked to several human health benefits [11]. The addition of flaxseeds in dairy cows’ nutrition may, to some extent, meet consumer demand for functional foods through increased PUFA, MUFA, and n-3 and decreased SFA levels [3]. However, information on the effects of other leguminous feedstuffs, including lupin seed or combinations of flaxseed with variable feedstuffs, on milk production and cheese composition and organoleptic characteristics, are scarce. A mixture of flaxseed and lupins in dairy cows’ diets can provide a well-balanced diet; however, to the best of our knowledge, there are no studies available in the literature addressing the effects of this combination on the characteristics of cow’s milk and white cheese.

Soybean meal has served as the primary protein source for dairy cows’ maintenance and production needs. At the same time, the broad concept of sustainability has led to an increased interest in searching for alternative feeds that could serve as a local and effective feed source with a lower environmental footprint [12]. In the current study, 50% soybean meal in the diet of dairy cows was replaced by equal quantities of flaxseed and lupin seeds. The effects of this dietary substitution were investigated on milk production and composition, as well as on the nutritional value and organoleptic characteristics of white cheese. The hypothesis of this study was that the addition of flaxseed and lupins in cows’ rations will maintain milk yield along with an increased content of unsaturated fatty acids in milk and white cheese and enhanced antioxidant status.

## 2. Materials and Methods

### 2.1. Ethics Guidelines of the Animal Research

All animals received humane care. The procedures described herein were according to the principles and guidelines of the European Union regulations and were approved by the Ethical Committee of Animal Welfare of the University of Thessaly (license number 112/19/9/20). Animals were routinely vaccinated and their health was monitored by a veterinarian who visited the farm at least once per week. 

### 2.2. Experimental Design and Dietary Treatments 

During the coldest period of the year (November to March), 60 Holstein cows (30 multiparous and 30 primiparous) were randomly selected and housed in a commercial dairy farm located in Thessaly, Greece. The cows were allocated into two equally sized treatment groups, with 30 cows each (15 multiparous and 15 primiparous). The animals of each group were housed in 5 separate pens, with 6 cows each, forming 5 different subgroups. Each group and subgroup were balanced for live weight (L.W.), age, and body condition. An adaptation period of two weeks, prior to the initiation of the experiment, was used for the animals to be accustomed to the modified ration. The experiment started 21 days before the expected calving and it was lasted until day 40 of the lactation (total duration of 60 days). The general and uterine health of the postpartum animals was evaluated every week until the third week postpartum and every third week thereafter. The control group (CO) was fed the basal ration with corn silage, lucerne hay, and wheat straw and a concentrate containing maize, soybean meal, and sunflower meal as the primary constituents (Table 1). In the diet of the second group (FL), 1.4 kg of soybean meal, corresponding to 50% of the total soybean meal participating in the basal ratio, was replaced by 1.2 kg of flaxseed and 1.2 kg of lupin seed. The control group (CO) consisted of multiparous cows (*n* = 15) with mean parity 3.1 ± 1.4 and BCS 3.2 ± 0.5 and heifers (*n* = 15) with BCS 3.1 ± 0.4; the FL groups consisted of multiparous cows (*n* = 15) with mean parity 3.3 ± 0.8 and BCS 3.3 ± 0.2 and heifers (*n* = 15) with BCS 3.1 ± 0.3. Both lupin seed and flax seed were cultivated exclusively for the experiment in the area of Imathia, Macedonia, North Greece. The white lupin (*Lupinus albus*, *Leguminosae*) variety used, “Multitalia”, is draft resistant, with a reduced number of alkaloids and a protein content of 35%, while the flaxseed (*Linum usitatissimum*) variety used, “Galaad”, has a protein content of 25%. Rationing was accomplished by following the recommendations of INRA [13], and both diets were isoenergetic and isonitrogenous (Table 1). A constant feed allowance was offered in both groups before and after calving. The animals were fed twice daily, using a feed mixer wagon; to estimate feed consumption, feed residuals were collected and weighed prior to feed provision. Fresh water was available ad libitum. Cows were milked mechanically thrice daily at 5:30, 12:30, and 19:30, in a 24-parallel milking parlor with individual ear-tag identification. Milk production was recorded at each milking. Environmental data were recorded using an automated electronic sensor that monitored temperature humidity index at real time (CMP Impianti, Brescia, Italy). Three animals, two mature cows that were diagnosed with clinical mastitis (one in each group) and one heifer from the FL group with left abomasal displacement, were excluded from the study.

### 2.3. Sample Collection

Feed samples were collected monthly and analysed for dry matter, crude protein, ether extract, and ash according to AOAC [14], and neutral detergent fiber according to the method of Van Soest and McQueen [15]. Further analysis was performed on feed samples to determine FA profiles [16]. Feed residuals were collected and considered in the calculations. Each week, individual milk samples of 200 mL were collected daily from each cow and mixed to a final weekly sample (210 mL). This sample was analysed for chemical composition, FAs profile, and oxidative status determination. For the white cheese production, bulk milk was collected from all daily milkings for each experimental group at days 30 and 60. White cheeses were analysed for chemical composition, fatty acid profile, lipid, and protein oxidation status and organoleptic properties.

### 2.4. Determination of Milk Components 

The composition of raw milk samples was analysed using the calibrated MilkoScan 4000 (FOSS Electric, Integrated Milk Testing^TM^, Hillerod, Denmark), providing increased accuracy. Each individual sample was tested for the following variables: protein, fat, lactose, solid non-fat (SNF), and total viable bacteria counts. The Fossomatic 5000 Basic (FOSS Electric, Hillerod, Denmark) instrument was used to determine milk somatic cell counts (SCC). 

### 2.5. White Cheese Production and Determination of Its Physicochemical and Organoleptic Characteristics

The cheesemaking process was carried out in a pilot cheesevat (ALPMA KBA 400, Alpma Alpenland Maschinenbau, Rott am Inn, Germany) of 400 L milk, separated in two cheesevats of 200 L each. Thus, every cheesemaking trial involved 200 L of CO milk and 200 L of FL milk, keeping all the other parameters the same and checking the difference in the yield. This cheesemaking trial was repeated 5 times. The differences in the cheese yield were not significant.

Fresh cow milk was normalised to a fat to protein ratio of 1.2, pasteurised at 73 °C for 15 s (GEA pilot plant, GEA Test Center for Aseptic Processing and Filling located at Ahaus, Germany), and then cooled to 35 °C. Cheese manufacturing was carried out according to the traditional standard procedure for Telemes cheese. Briefly, calcium chloride solution (0.02% *w*/*v*) was incorporated into the milk and a blend of thermophilic and mesophilic starter culture was inoculated to the milk 10 min before the addition of rennet. The rennet (Fermented Produced Chymosin, 600 IMCUs) was added to induce coagulation in approximately 50 min at 35 °C. After coagulation, the curd was cut into cubes of 2 cm and left to rest for 30 min. The sliced curd was then transferred into molds and stored in the drainage room at 18 °C for 24 h. During this period the molds are turned upside down several times to complete whey drainage. Subsequently, the cheeses were placed in containers with brine (8°Be) and kept at 18 °C for approximately 10 days for cheese ripening. Afterwards, the product was placed in specified containers filled with brine (8°Be) and preserved at 18 °C for almost two weeks to complete the process of cheese ripening. Afterwards, the containers were kept at 4 °C for the entire ripening process; this lasted two months. Following the ripening period, cheese samples were collected to perform physicochemical and organoleptic analyses. The pH of the cheeses was assessed using a pH meter (GLP-21, CRISON Instruments SA., Barcelona, Spain). The determination of moisture was made in accordance with the sea sand method [17]. Sodium chloride content was determined according to the standard method of the International Dairy Federation [18]. Lastly, total nitrogen (TN) was estimated using the Kjeldahl method [19] and fat content was evaluated using the Gerber van Gulik method [20].

The sensory panel was composed of 14 participants aged from 22 to 65 years. The panelists were requested to record their opinion on the flavor, appearance, and body and texture of 12 cheese samples from each group, cut into small cubes with a 2 cm^3^ mass. All parameters were set up on a hedonic scale from 1 (negative perception) to 9 (positive perception). The scale from 1 to 3 corresponded with low level of acceptance, from 4 to 6 with acceptance, and from 7 to 9 with very high acceptance. Panelists were also asked to record supplementary comments if necessary. Each participant provided scores for all samples from both treatment groups.

### 2.6. Determination of Milk and White Cheese Fatty Acids Profile

Individual milk and cheese samples were analysed for the determination of their FA profile. Fatty acid methyl esters (FAMEs) were evaluated in accordance with the Bligh and Dyer [21] method and the International Organisation for Standardisation ISO [22], as reported by Papaloukas et al. [23]. 

### 2.7. Determination of TBARS and Protein Carbonyls in Milk and White Cheese 

For the thiobarbituric acid reactive substances (TBARS determination), an aliquot of 100 μL of milk or cheese sample was mixed with 500 μL of trichloroacetic acid (TCA) solution (35% *w*/*v*) and 500 μL of Tris–HCl (200 mmol/L; pH 7.4), and incubated for 10 min at 20 °C. Then, 1 mL of Na_2_SO_4_ and 55 mM thiobarbituric acid solution were added, and the samples were incubated at 95 °C for 45 min. Finally, the absorbance was measured at 530 nm using a spectrophotometer (UV-1700 PharmaSpec, Shimadzu, Japan). Results were expressed as ng MDA per l ml of milk or cheese. Protein carbonyls assessment was applied in both milk and cheese samples following the method described in detail by Patsoukis et al. [24].

### 2.8. Determination of Total Phenolic Content and Interactions with DPPH of Experimental Diets, Milk, and Cheese Samples

Feed, milk, and cheese samples of the experimental groups were also analysed for their total phenolic content according to the method described by Singleton et al. [25], using the Folin–Ciocalteu assay, and expressed as gallic acid equivalents (GAE). 

The DPPH (1.1-diphenyl-2-picrylhydrazyl) activity of the feed, milk, and cheese samples was determined with respect to hydrogen-donating or radical-scavenging ability, according to the method of Peperidou et al. [26].

### 2.9. Statistical Analysis

Data were analysed by ANOVA using SPSS version 25 (SPSS Inc., Chicago, IL, USA) statistical package. The animals were separated into 2 groups with 5 subgroups/pens. Each pen was set as the experimental unit. The homogeneity of the variances was tested using ANOVA (Tukey’s or Duncan’s post-hoc test) test. A *p*-value less than 0.05 was considered statistically significant.

## 3. Results and Discussion

The objective of this study was to investigate whether the dietary addition of flaxseed and lupins would improve milk and white cheese composition and FAs, as well as the organoleptic characteristics of white cheese. Dairy food products derived from cow’s milk, especially cheeses, are highly rated in the preference of consumers; however, recent concerns have arisen due to the fact that milk has been labelled as a food rich in saturated fatty acids with values that can reach even 70% of total fatty acid content. Moreover, an increase of polyunsaturated fatty acids would make milk or cheese more prone to lipid or even protein oxidation. In the present experiment, the diet GAE content of the CO and FL groups was 0.19 mg GAE per g of dry matter and 0.22 mg GAE per g of dry matter, respectively. We found that DPPH and total phenolic antioxidant values were similar among the different groups (Table 2). 

No difference was recorded in the overall health status between the animal groups. Milk production was not affected by dietary treatments (Figure 1), neither in primiparous nor in multiparous cows. Although the FL ration was formed based on soybean meal partial substitution, diets were kept isonitrogenous and isocaloric among the experimental groups (Table 1). We can assume that energy utilisation towards milk synthesis remained similar and thus animals characterised by the same genetic potential were able to maintain their production capabilities. Santillo et al. [3] made similar observations after investigating the effects of whole flaxseed supplementation on diary cows’ milk production, although a small numerical decrease in the flaxseed group was mentioned. 

Milk protein and fat content are generally affected by fundamental influential factors, including animal genetics, feed intake, and ingredient composition [27]. In our study, feed composition differed among the groups, yet milk protein and fat content fluctuated within the same range among control and treated cows (Table 3). Caroprese et al. [5] attributed cows’ milk protein increase, after the inclusion of whole flaxseed in their diet, to the N boosted flow towards the duodenum. Flaxseed proteins escape partial rumen degradation; thus, higher amounts of essential amino acids are available in the udder for protein synthesis. Our experiment was based on partial substitution of soybean meal by flaxseed and lupin. Although both flaxseed and lupin seeds have adequate protein content, soybean meal has a superior protein content, providing higher duodenal flow of both PDIA and PDII [13]. In terms of milk fat content, our results are in agreement with Petit et al. [28], who reported similar values among supplemented and control cows. Milk SCC were not affected by oil seeds supplementation, similar to the results reported by Martin et al. [29]. In a previous study [30], we found that flaxeed and lupin seed contribute to enhanced cellular immunity in the endometrium and in divergence in acute phase proteins concentrations; these are non-specific innate immune components. However, the immunomodulatory effects of FL failed to affect the SCC in both the present and the previous study. 

Flaxseed and lupin incorporation in the FL group significantly influenced the milk fatty acid profile (Table 4), confirming previous findings investigating the use of oilseeds in dairy cows’ ration as a method to produce high quality functional milk [3,31,32]. According to our findings, the FL group showed an increase in SFAs; in addition, there was a significant increase in PUFA content when compared to the CO (*p* < 0.05). Persistent participation of the saturated FAs in human nutrition may act as a predisposing factor for several cardiovascular diseases [33]. On the contrary, PUFAs are linked with several cardiovascular benefits. C18:3n-3 FA was significantly abundant in milk samples from the FL group when compared with CO group (*p* < 0.05). Analogous findings were reported after the supplementation of extruded linseed in dairy cows’ diets [34].

Researchers are investigating the effects of C18:2 cis-9, trans-11, which is a conjugated linoleic acid isomer acknowledged as a human health promoter [35]. Despite the fact that only minor concentrations are received with feedstuffs, this fatty acid, naturally found in dairy products, can be synthesised either in the rumen through omega-6 fatty acids biohydrogenation or in the mammary gland after C18:1 trans-11 desaturation by Δ9-desaturase. Flaxseed and lupin seed, in a moderated degree, have rich oil content, abundant in linolenic acid. As such, both seeds dietary inclusiong represents a feasible method to increase ruminant milk CLA content, by supporting the above-described mechanisms [9,36]. Several studies focusing on milk fatty acid profile enhancement have shown that flaxseed supplementation leads to increased milk CLA concentrations [37,38]. 

Cheese fatty acid profile was found to have significant differences that can be attributed to the different dietary treatments (Table 5). Cheese content showed increased n-3 FAs (*p* < 0.05) and PUFA (*p* < 0.05) in the FL group. Milk fat has been identified as the primary aspect affecting cheese fat chemical profile, while cheese manufacturing methods have a smaller influence [39]. Our results suggest that oilseeds rich in polyunsaturated fatty acids have the ability to beneficially alter white cheese nutritional aspects, leading to the production of a product adjusted to health recommendations. 

The chemical analysis of white cheese did not differ between the two groups (*p* > 0.05) (Table 6). Additionally, cows fed diets with flaxseed and lupins showed compositional and organoleptic properties quite similar to control cheese (Figure 2). Appearance, texture, and flavor were acceptable and desirable in both cheeses, with scores above 6 in a scale of 1 to 9. White cheese is manufactured in Greece and several Balkan countries as part of their traditional diet; however, in the last decades, a significant production of white cow cheese (like Telemes) has also been produced in several EU countries, and is well accepted by European consumers. Subsequently, cheese ripening can affect cheese composition as maturity may interact with internal enzymes or the production of acidic components [40]. However, it seems that energy and protein levels are major determinants of milk production for cows with the same genetic potential. Holstein dairy cows show a good adaptation to climates all over the world, except for hot tropical climates, and provide high milk yield with constant characteristics. 

An additional aim of this study was to evaluate whether a dietary mixture of flaxseed and lupins would affect the antioxidant status of milk and cheese. This is because an increase in PUFAs could make milk more vulnerable to oxidation [41]. Evidentially, diet supplementation with flaxseed and lupins did not deteriorate lipid oxidative stability in milk and cheese samples (Table 7). In addition, both milk and cheese samples from the FL group disaplyed lower protein carbonyls values, providing evidence that those animals could better cope with oxidation pathways, showing similar values with the animals fed the control diet. Correspondingly, the FL diet showed significanlty increased total phenolics; these have been found to contain antioxidant properties through their ability to react with free peroxyl radicals [42]. Future research efforts should focus on defining specific types and composition, along with optimum inclusion rates, in order to optimise their level of the favorable and consistent responses, and thus to effectively utilise their benefits. 

In summary, the role of flax and lupin can serve multiple purposes: (a) to reduce the milk production cost by limiting the inclusion rate of the expensive and imported (to most Mediterranean countries) soybean meal, (b) to add increased value to degraded fields, increasing farmers’ income, and (c) to allow production of functional dairy products to conform with modern consumer demands.

## 4. Conclusions

The substitution of soybean meal by flaxseed and lupins in the diets of Holstein cows sustained high milk yield and improved omega-3 content in the milk and cheese from treated cows. Milk and cheese fat oxidation was preserved in low levels regardless of the higher PUFA content present in the treated diet. The results of the present study warrant further investigation towards the greater reduction or even elimination of soymeal from dairy cows’ diets to produce novel products or to improve the desirable characteristics of dairy products.

## Figures and Tables

**Figure 1 animals-13-01159-f001:**
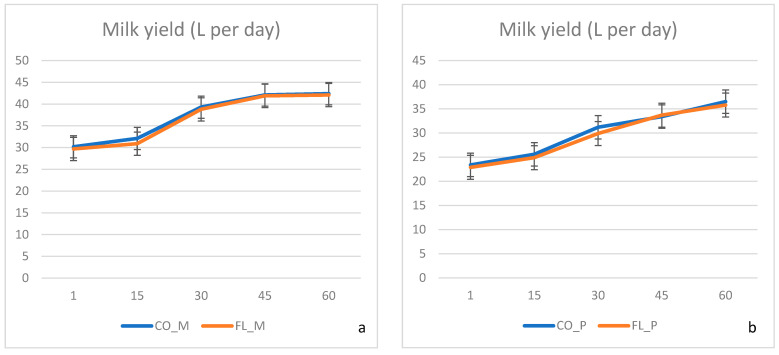
(**a**,**b**) Mean milk yield during the experimental period 1–60 days postpartum. CO-M: control multiparous cows, FL-M: flaxseed and lupin treated multiparous cows, CO-P: control primiparous cows, FL-P: flaxseed and lupin treated primiparous cows.

**Figure 2 animals-13-01159-f002:**
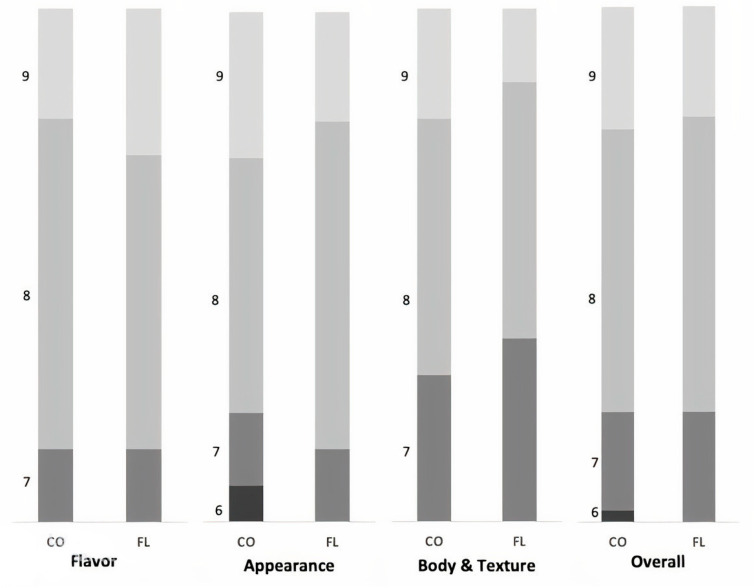
Effect of dietary addition of flaxseed and lupin on sensory panel scores of cheese samples on flavor, appearance, body, and texture and overall. N° replicates: 24 (12 samples per treatment) scored by 14 panelists. Results range: 1–3 low acceptance, 4–6 sufficient acceptance, and 7–9 very high acceptance.

**Table 1 animals-13-01159-t001:** Daily ingredient allowance and chemical composition of the TMR offered to dairy cows during the experimental period.

	Diet ^1^	
Item	CO	FL
Corn silage (kg)	23	23
Alfalfa hay (kg)	2.7	2.7
Wheat straw (kg)	2.3	2.3
Corn (kg)	3	2
Barley (kg)	1.2	1
Soymeal (CP: 47%) (kg)	2.8	1.4
Rapeseed meal (CP: 32%) (kg)	2.1	2
Molasses (kg)	2.5	2
White Lupin (kg)	-	1.2
Flaxseed (kg)	-	1.2
Total	39.6	38.8
Chemical analysis		
DM %	60.00	57.00
DM (g)	23.200	23.109
CP % DM	15.85	15.91
ADF % DM	18.52	22.20
NDF % DM	37.69	41.25
Ether extract % DM	7.55	10.4
UFL/kg	0.501	0.499
Starch % DM	20.8	19.2
Sugars % DM	8.9	7.9
PDIN g/kg DM	2417.5	2003
PDIE g/kg DM	2143.5	1738
PDI g/kg DM	1987	1810
PDIN/UFL	127.5	110.7
PDIE/UFL	113.11	96
RPB	506	530
Ca % DM	0.33	0.31
P % DM	0.33	0.3
Ca, absorb(g)	28.1	27.4
P, absorb(g)	24.7	25.1
Mg % DM	0.19	0.18
Na % DM	0.10	0.09
S % DM	0.28	0.25
Fatty acids g/kg DM		
Lauric-C12	3.25	3.24
Myristic-C14	3.84	3.74
Palmitic-C16	252.5	255.03
Palmitoleic-C16-1n7	4.9	4.98
Stearic-C18	4.93	3.34
Oleic-C18-1n9	23.3	34.66
Linoleic-C18-2n6	50.9	55.9
A-linolenic-C18-3n3	112.6	152.34

^1^ CO = Control group; FL = Flaxseed and lupin group.

**Table 2 animals-13-01159-t002:** Feed DPPH and total phenolic content.

Item	CO ^1^	FL
DPPH ^2^ (20 mg/mL)		
20 min	30.21	28.14
60 min	29.15	27.33
TP	15.05	16.04

^1^ CO = Control group; FL = Flaxseed and lupin group; ^2^ DPPH = 2,2-Diphenyl-1-picrylhydrazyl.

**Table 3 animals-13-01159-t003:** Milk fat, protein, SCC, and CFU of cows fed the experimental diets ^1^.

	Treatment ^1^			
Item	CO-M	FL-M	CO-P	FL-P
Mean fat (%)	4.01 ± 0.31	3.96 ± 0.52	3.88 ± 0.34	3.79 ± 0.57
Mean protein (%)	3.31 ± 0.12	3.40 ± 0.21	3.42 ± 0.31	3.37 ± 0.22
Mean SCC ^2^ (×1000)	327 ± 121	352 ± 94	307 ± 111	276 ± 74
Mean CFU ^3^ (×1000)	7.5 ± 1.2	8.1 ± 1.0	8.9 ± 0.9	9.4 ± 1.4

^1^ CO-M: control multiparous cows, FL-M: flaxseed and lupin treated multiparous cows, CO-P: control primiparous cows, FL-P: flaxseed and lupin treated primiparous cows. ^2^ SCC: somatic cell count. ^3^ CFU: Colony forming units.

**Table 4 animals-13-01159-t004:** Fatty acid composition (%) of milk from cows subjected to different diets.

	Treatments ^1,2^		
Fatty Acid	CO	FL	*p* Value ^3,4^
Caproic-C6:0	0.59 ± 0.23 ^b^	1.19 ± 0.44 ^a^	<0.05
Caprylic-C8:0	0.61 ± 0.28 ^b^	1.89 ± 0.20 ^a^	<0.05
Capric-C10:0	1.13 ± 0.74 ^b^	6.87 ± 0.38 ^a^	<0.05
Lauric-C12:0	1.21 ± 0.58 ^b^	4.51 ± 0.49 ^a^	<0.05
Tridecylic-C13:0	1.67 ± 0.84 ^b^	2.25 ± 0.70 ^a^	<0.05
Myristic-C14:0	5.64 ± 1.89 ^b^	11.57 ± 0.30 ^a^	<0.05
Myristoleic-C14:1ω5	0.69 ± 0.09 ^a^	0.42 ± 0.04 ^b^	<0.05
Pentadecylic-C15:0	0.58 ± 0.32 ^b^	0.98 ± 0.21 ^a^	<0.05
cis-10-Pentadecenoic-C15:1ω5	0.26 ± 0.21	0.42 ± 0.09	NS
Palmitic-C16:0	28.27 ± 1.35	27.71 ± 0.54	NS
Palmitoleic-C16:1ω7	3.10 ± 0.94 ^a^	0.80 ± 0.13 ^b^	<0.05
cis-7-Hexadecenoic-C16:1ω9	0.44 ± 0.08	0.44 ± 0.08	NS
cis-10-Heptadecenoic-C17:1ω7	0.59 ± 0.09 ^a^	0.27 ± 0.03 ^b^	<0.05
Stearic-C18:0	12.18 ± 2.01 ^a^	9.51 ± 1.07 ^b^	<0.05
Oleic-C18:1ω9c	39.56 ± 0.48 ^a^	23.02 ± 0.93 ^b^	<0.05
Cis-Vaccenc-C18:1ω7c	0.61 ± 0.18 ^b^	1.55 ± 0.16 ^a^	<0.05
Linoleic-C18:2ω6c	1.96 ± 0.26 ^b^	3.29 ± 0.18 ^a^	<0.05
γ-Linoleic-C18:3ω6	0.32 ± 0.09 ^b^	1.44 ± 0.17 ^a^	<0.05
α-Linoleic-C18:3ω3	0.30 ± 0.18 ^b^	1.20 ± 0.13 ^a^	<0.05
Arachidonic-C20:4ω6	0.29 ± 0.26 ^b^	0.66 ± 0.35 ^a^	<0.05
∑SFA ^5^	51.87 ± 4.78 ^b^	66.49 ± 1.11 ^a^	<0.05
∑UFA ^6^	48.13 ± 4.78 ^a^	33.51 ± 1.11 ^b^	<0.05
∑MUFA ^7^	45.25 ± 5.23 ^a^	26.91 ± 1.05 ^b^	<0.05
∑PUFA ^8^	2.88 ± 0.56 ^b^	6.60 ± 0.57 ^a^	<0.05

^1^ CO: control multiparous cows, FL: flaxseed and lupin treated multiparous cows. ^2^ Each value is the mean of triplicate determinations ± s.d. ^3^ Different lowercase letters within the same row indicate significant differences according to Duncan’s test (*p* ≤ 0.05). ^4^ NS: Not Significant (*p* > 0.05). ^5^ Saturated fatty acids. ^6^ Unsaturated fatty acids. ^7^ Monounsaturated fatty acids. ^8^ Polyunsaturated fatty acids.

**Table 5 animals-13-01159-t005:** Fatty acid composition (%) of white cheese from milk of cows subjected to different diets.

	Treatments ^1,2^		
Fatty Acid	CO	FL	*p* Value ^3,4^
Caproic-C6:0	0.77 ± 0.10 ^b^	1.29 ± 0.13 ^a^	<0.05
Caprylic-C8:0	0.58 ± 0.04 ^b^	1.76 ± 0.06 ^a^	<0.05
Capric-C10:0	1.55 ± 0.07 ^b^	6.83 ± 0.05 ^a^	<0.05
Lauric-C12:0	0.12 ± 0.02	0.27 ± 0.09	NS
Tridecylic-C13:0	2.13 ± 0.02 ^b^	4.82 ± 0.25 ^a^	<0.05
Myristic-C14:0	0.27 ± 0.05	0.37 ± 0.13	NS
Myristoleic-C14:1ω5	8.83 ± 0.04 ^b^	11.59 ± 0.08 ^a^	<0.05
Pentadecylic-C15:0	0.87 ± 0.01 ^a^	0.34 ± 0.02 ^b^	<0.05
cis-10-Pentadecenoic-C15:1ω5	0.90 ± 0.02	1.05 ± 0.07	NS
Palmitic-C16:0	0.90 ± 0.01 ^a^	0.26 ± 0.04 ^b^	<0.05
Palmitoleic-C16:1ω7	29.86 ± 0.09	27.31 ± 0.29	NS
cis-7-Hexadecenoic-C16:1ω9	2.23 ± 0.02 ^a^	0.93 ± 0.01 ^b^	<0.05
cis-10-Heptadecenoic-C17:1ω7	0.43 ± 0.03	0.37 ± 0.02	NS
Stearic-C18:0	0.47 ± 0.02	0.25 ± 0.02	NS
Oleic-C18:1ω9c	12.73 ± 0.05 ^a^	9.28 ± 0.14 ^b^	<0.05
Cis-Vaccenc-C18:1ω7c	0.27 ± 0.02 ^b^	1.18 ± 0.12 ^a^	<0.05
Linoleic-C18:2ω6c	2.27 ± 0.02 ^b^	3.88 ± 0.16 ^a^	<0.05
γ-Linoleic-C18:3ω6	0.46 ± 0.05 ^b^	1.32 ± 0.15 ^a^	<0.05
α-Linoleic-C18:3ω3	0.39 ± 0.05 ^b^	1.17 ± 0.08 ^a^	<0.05
Arachidonic-C20:4ω6	0.25 ± 0.01	0.19 ± 0.01	NS
∑SFA ^5^	57.54 ± 0.11 ^b^	64.87 ± 0.83 ^a^	<0.05
∑UFA ^6^	42.46 ± 0.11 ^a^	35.13 ± 0.83 ^b^	<0.05
∑MUFA ^7^	38.59 ± 0.03 ^a^	27.53 ± 0.74 ^b^	<0.05
∑PUFA ^8^	3.87 ± 0.11 ^b^	7.60 ± 0.21 ^a^	<0.05

^1^ CO: control multiparous cows, FL: flaxseed and lupin treated multiparous cows. ^2^ Each value is the mean of triplicate determinations ± s.d. ^3^ Different lowercase letters within the same row indicate significant differences according to Duncan’s test (*p* ≤ 0.05). ^4^ NS: Not Significant (*p* > 0.05). ^5^ Saturated fatty acids. ^6^ Unsaturated fatty acids. ^7^ Monounsaturated fatty acids. ^8^ Polyunsaturated fatty acids.

**Table 6 animals-13-01159-t006:** Cheese composition of cows fed the experimental diets ^1^.

	Treatment ^1^			
Item	CO	FL	SEM	*p* Value ^2^
Fat (%)	20.68	23.66	0.882	NS
Protein (%)	14.42	16.49	0.423	NS
Lactose (%)	1.39	0.15	0.456	NS
Moisture (%)	59.61	56.38	0.879	NS
Ash (%)	2.34	2.63	0.912	NS

^1^ CO: control multiparous cows, FL: flaxseed and lupin treated multiparous cows. ^2^ NS: Not Significant (*p* > 0.05).

**Table 7 animals-13-01159-t007:** Effects of dietary supplementation of flaxseed and lupin seed on antioxidant status of milk and white cheese.

		Treatments ^1^		
Parameter	CO	FL	SEM	*p* Value ^2^
	Milk			
TBARS ^3^ (ng/mL)	0.25	0.24	0.17	NS
Protein carbonyls (ng/mL)	22.89 ^b^	18.43 ^a^	0.68	<0.05
TP ^4^	0.04 ^b^	0.11 ^a^	0.06	<0.05
	White cheese			
TBARS (ng/mL)	3.12 ^a^	0.48 ^b^	0.48	<0.05
Protein carbonyls (ng/mL)	28.47 ^a^	11.49 ^b^	0.68	<0.05
TP	0.02 ^b^	0.09 ^a^	0.03	<0.05

^1^ CO: control multiparous cows, FL: flaxseed and lupin treated multiparous cows. ^2^ Different lowercase letters within the same row indicate significant differences according to Duncan’s test (*p* ≤ 0.05); NS: Not Significant (*p* > 0.05). ^3^ TBARS: thiobarbituric acid reactive substances. ^4^ TP: total phenolics.

## Data Availability

Not applicable.

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
