# Peer review of "Composition, Organoleptic Characteristics, Fatty Acid Profile and Oxidative Status of Cow’s Milk and White Cheese after Dietary Partial Replacement of Soybean Meal with Flaxseed and Lupin"

_animals, 2023, doi:10.3390/ani13071159_

Round 1

Reviewer 1 Report

Please see the document.

Author Response

Dear Reviewer,

We thank you for you comments!

We have made the revisions needed.

Reviewer 2 Report

Revisions required

Line 46: "Milk of FL had decreased palmitic and myristic, increased (p<0.05) oleic and (p<0.05) linolenic acid" the significances should be listed after the names of fatty acids. The same applies to a sentence later - significances should only be given after the values they relate to.

Please elaborate on the feeding process - time and method of feeding (feed mixer truck/wagon, robotic feeding system, or similar?)

Line 156: In the methodology, specify what parameters were monitored under "oxidation status".

Line 190: I would suggest moving part of methodology 2.5 "Determination of milk chemical composition" before part 2.4

Line 207: The abbreviation "TBARS" is not explained.

I think that for fatty acids and for chemical compounds subscripts should be used throughout the manuscript - instead of "Na2SO4" this "Na2SO4". Similarly, on line 223 there should probably be superscripts for "M-1 cm-1".

Line 243: The abbreviation "DPPH" should be explained at first use in the text.

Line 247: The abbreviation "DMSO" is not explained. However, it is only used once so maybe full name instead of abbreviation would be better.

Line 268: I would prefer this sentence to be rewritten to improve readability as DM is not used elsewhere in the text: 0.19 mg per g of dry matter and 0.22 mg GAE per g of dry matter. Also, it should be explained under the tables.

Line 276-277: The mention of discarded animals should instead be in the methodology.

Inconclusive/insignificant results should be simply described as inconclusive and there is no need to state "P>0.05", such as on lines 278 and 295.

Figures 1a and 1b could be merged - in any case, the graphs lack the expression of the units in which the production is shown, and it would be appropriate to adjust the range of the "y" axis to highlight the differences better

In Table 3, it is probably LSM and SELSM calculated by one-factor ANOVA. However, it is not very clear from the text for this table and should be better explained. Same for table 4.

In addition, there are erroneous spaces in table 3 (sometimes missing), and the description of "SCC" and "CFU" is also missing under the table.

Please unify the marking of evidence - somewhere it is "P<0.05" (in the results) and somewhere "p<0.05" (in the abstract and methodology). I believe it correctly should be p < 0.05 for publication in Animals journal.

Table 4 and 5 – The use of numerical indexes is a bit confusing in this table. Please edit to make it more understandable. Also, the 6 is missing in table footer.

In Table 5, it is necessary to recheck the letters indicating significance - for example, for 12:0, there is an index "b" for both values, and at the same time, it is stated as significant.

Line 335: The word "environment" is crossed out.

Line 369: incorrectly numbered table - it should be #6 here.

In Table 6, these are arithmetic means, or they are also values calculated using ANOVA. If they are LSMs, why are not SELSMs also stated?

Under Table 7, it is necessary to explain what "TBARS" and "TP" are. In this table, the exact "P" values are given, even though the methodology states that the "P<0.05" level will be used. Please edit.

For all tables, the letters indicating significance should go from left to right, or the designation "a" always for a higher value and "b" for a lower value - it is different in different tables. Please edit.

Please recheck the formatting in the "References" section. There are some author names listed as "DE LAITERIE" and others as "Patsoukis". The second way is correct.

Author Response

(The authors gave the same response as above.)
